# An Integrated Coastal Sediment Management Plan: The Example of the Tuscany Region (Italy)

**Enzo Pranzini [1], Irene Cinelli [1], Luigi E. Cipriani [2] and Giorgio Anfuso [3,*]**

[1]  Department of Earth Science, University of Florence, 50121 Firenze, Italy; enzo.pranzini@unifi.it (E.P.); irene.cinelli@hotmail.it (I.C.)
[2]  Direction of Soil Defense and Civil Protection, Region of Tuscany, Region of Tuscany, 50127 Firenze, Italy; luigi.cipriani@regione.toscana.it
[3]  Department of Earth Science, Faculty of Marine and Environmental Sciences, University of Cadiz, CASEM, 11510 Puerto Real, Cádiz, Spain
*   Correspondence: giorgio.anfuso@uca.es

**Abstract:** This paper presents the results of a study carried out to support the Region of Tuscany Coastal Sediment Management Plan, with the main aim of establishing the sediment budget considering the time span from 1981–1985 to 2005 for the 56 coastal sectors into which the 215 km-long continental sandy coast of Tuscany (Italy) was divided. The sand stability (according to a stability index) and colour compatibility (according to the CIEL*a*b* colour space with an acceptability range conforming to national guidelines) were determined in order to assess the possibility of using the available sediment in accreting sectors to nourish the beach in eroding areas. Only in two cases—i.e., the updrift of a harbour (at Viareggio) and in a convergence zone (at Marina di Pietrasanta)—are the volumes of sufficient magnitude to support a large nourishment project; however, the mean sand size is too small to guarantee efficient nourishment, even with medium-term stability. In contrast, the colour difference, in most of the cases, was shown to be acceptable. Other small sediment stocks, suitable for colour but not for grain size, can be used for periodic ephemeral nourishment works to support seasonal tourist activities. The limited resources available make it necessary to adopt a plan for their optimal use from a regional perspective. This kind of study is of great interest for the proposal of sound management actions to counteract the increasing erosion processes linked to climate change phenomena and human effects on rivers and coastal systems.

**Keywords:** key coastal erosion; sediment budget; fill stability; colour compatibility; beach nourishment

---

## 1. Introduction

Tourism is one of the most important industries in the world; global international tourist arrivals grew by 3.9% (1235 million people) in 2016 and 7% (1326 million people) in 2017 [1,2]. Beaches make up a major part of this market [3,4], especially along the Mediterranean coast, which is characterized by mild temperatures associated with annual precipitation in winter and a hot, dry season in summer, which is very attractive for "the Sun, Sea and Sand (3S) tourism" market [5].

Beach erosion, which is a relevant threat to 3S tourism [6], is the result of a deficit in the coastal sediment budget due to the prevalence of outputs over inputs. This process can be locally countered by reducing debits—i.e., sand loss due to erosion processes—by putting shore protection structures in place; however, these increase the deficit in downdrift coastal sectors [7]. Input credits can be increased, favouring soil erosion and river transport or soft rock cliff erosion; both strategies are poorly-suited to developed areas, such as the Tuscan coast in north-western Italy.

The direct injection of sediment quarried on land or extracted from the sea floor to the coast, by means of artificial nourishment works, is the most used approach to increase sediment input [6,8,9].

However, coastal segments where the sediment budget is positive exist; they are observed in both natural conditions, e.g., when sediments are accumulated near headlands at the end of coastal cells or in longshore transport convergence zones [10], and in human-created conditions, i.e., updrift, or in correspondence with coastal structures such as harbours, jetties and breakwaters [11,12].

The sediment circulation framework and sediment budget at the regional scale have to be characterized [13–16] based on the sediment volume continuity equation, which can be calculated both in an analytical way, e.g., with empirically-derived equations [13,17–19] by propagating waves in an area with well-known coastal morphologic and sedimentological characteristics, or by measuring eroded and accumulated sediment volumes, with the latter being mostly relevant to coastal structures [20] or at converging cells limits [13,21]. The former method considers "sediment potential transport", and is easily applicable when nearshore morphological, sedimentological and wave climate data are available [22], whereas the latter deals with actual sediments, but needs accurate historical topographic surveys of the emerged and submerged beach [12,23].

The importance of coastal sediments as a resource for the sound management of the coastal environment and for countering erosion and the effects of climate change has increasingly taken on a strategic value in recent years [24]. This is especially true for Mediterranean coastal areas, which are strongly suited to tourism, and are often characterized by excellent environmental and cultural value, which are strong drivers of economic growth. In this scenario, the management of sediment resources entails the need for specific regional policies for the protection and sustainable use of the different sources and of strategic reservoirs, as underlined by the "EUROSION" project recommendations [25], and by the indications of the "National Guidelines for Coastal Erosion in Italy" [26]. The different natures of the diverse sources involved result in the need to identify adequate management, regulatory and authorization systems in order to favour the sustainable use of sediment resources in a complementary and synergistic way for the purposes of coastal management and protection, along with the principles of the Protocol on Integrated Coastal Zone Management in the Mediterranean [27] for sustainable growth combined with the objectives of territorial safety and the protection of the coastal environment.

A sound regional sediment budget can address shore protection strategies based on sediment bypass from accreting to eroding sectors, which is being carried out though a Regional Sediment Management (RSM) Project developed to implement adaptive management strategies across multiple projects, optimizing the use of sediment while supporting sustainable solutions to navigation and dredging, flood and storm damage reduction, and environmental enhancement missions [16,28,29]. Particular attention also has to be devoted to the effects on beach stability of the extraction of sediments from the nearshore [30].

The coast of Tuscany has experienced significant erosion rates in past decades (Figure 1). To counteract this trend, the Regional Administration planned to quantify and characterize the sediment availability in the coastal area, consisting mainly of sand accumulated updrift of major ports and marinas, and which could allow for significant bypass interventions to nourish eroding sectors. In parallel, sand stocks on the shoreface were analysed which could be used to carry out seasonal beach nourishments to support seaside tourism activities, which represent a significant part of the regional economy. In order to quantify these resources, characterize sediments and assess their suitability for the different coastal sectors experiencing shoreline retreat, a project founded by the Region of Tuscany was carried out at the University of Florence in 2015.

Several aspects have to be taken into account to properly develop a management plan; e.g., designing an adequate, stable artificial beach profile [31–35] and choosing how to suitably borrow sediment, which has to be compatible with the natural one according to textural [23,36] and chromatic characteristics [37–39]. These two aspects have been analysed in this paper.

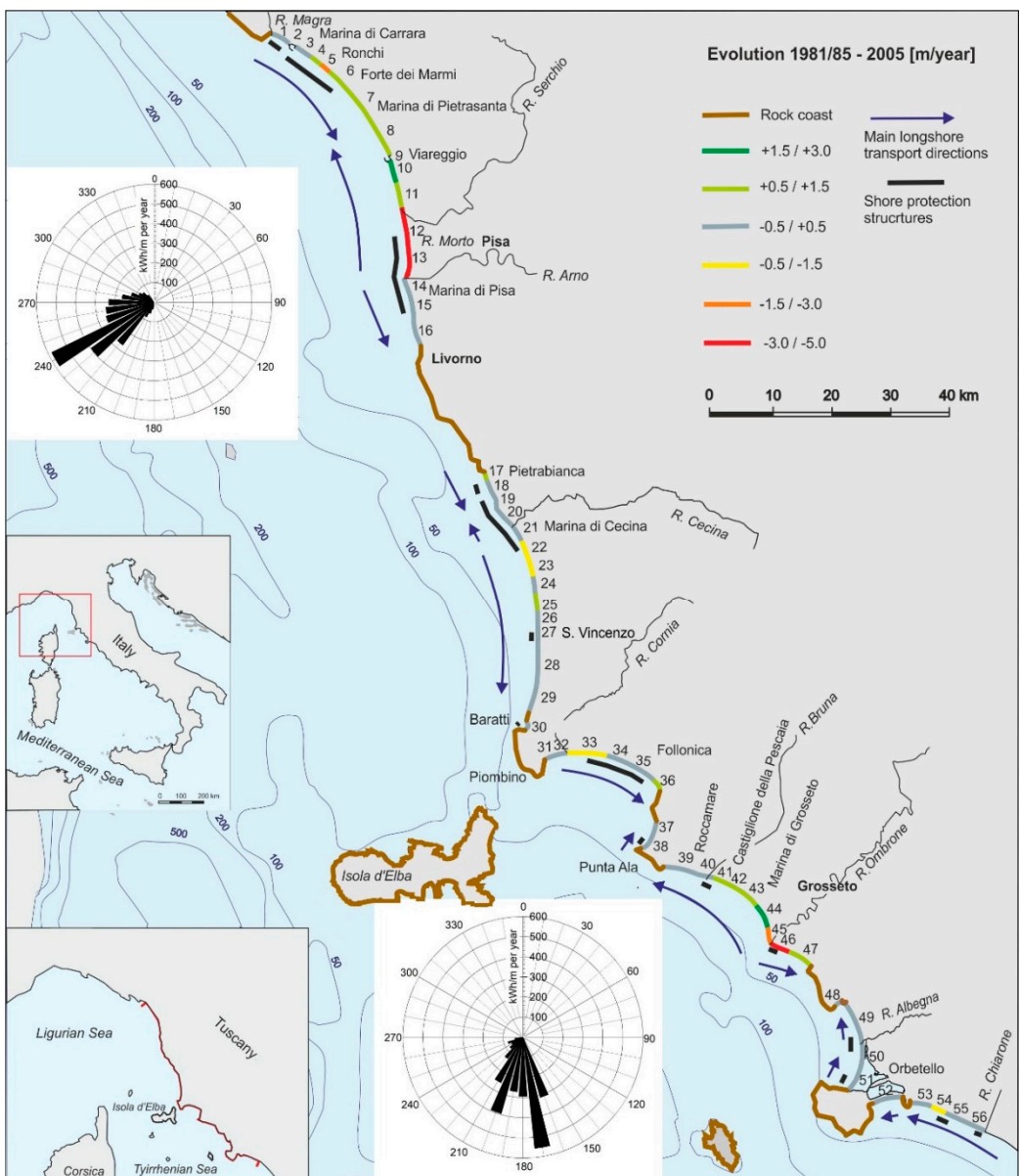

**Figure 1.** Shoreline evolution along the Tuscan coast from 1981–1985 to 2005; the sectors into which the coast has been divided for the present study are also shown. In Appendix A, the time interval for each coastal sector is presented. Wave energy (KWh/m) directional distribution roses obtained from KNMI data are also reported.

## 2. Study Area

The continental coast of the Tuscany Region, a microtidal environment, is approximately 380 km long, 215 km of which comprises sandy beaches, with few segments characterized by mixed sand and gravel sediments. This calculation excludes 9 km, or 4.0% of the total coastline, which once consisted of sand beaches but today is protected by seawalls, usually emplaced in correspondence of residential areas and coastal roads, or hosts port facilities [40,41].

The longest continuous offshore wave data series for this area is given by the Koninklijk Nederlands Meteorologisch Instituut (KNMI); wave energy (KWh/m) directional distribution roses derived from 1961–1990 wave data are reported for the areas north and south of Elba Island (Figure 1). In the northern area, wave energy essentially comes from south-west, and in the southern one, from the south.

For more than a century, the Tuscan coast (Figure 1) has experienced an erosive process that, despite the implementation of various counter measures, continues to expand [41]. The coastal evolution in Tuscany was obtained by comparing two shorelines: from1981–1985 and from 2005 [42]. The 1981–1985 shoreline was obtained by means of aerial photographs at an original scale of 1:13,000 with an accuracy of 5 m [40], and the 2005 shoreline was obtained by means of accurate DGPS surveys.

Coastal erosion/accretion rates (m/yr) for the investigated period are shown in Figure 1; the presented values comprise mean shoreline changes in 821 segments, each of which is approximately 250 m long. At each segment, the mean shoreline displacement value (m) was computed by dividing beach area variation by segment length [43] using Geographic Information System (QGIS) tools.

In the period from 1981–1985 to 2005, 9.1% of the regional beaches underwent sever erosion, with retreat rates of more than 3 m/yr; another 12.0% suffered less intense erosion, where beach retreat was between 0.5 and 3.0 m/yr; 27.0% had a shoreline retreat that was slower than 0.5 m/yr (Figure 2). This latter amount, as well as that corresponding to slow (<0.5 m/yr)-accreting areas (+21.8%) were within the mapping accuracy and intrinsic beach variability, and hence have been considered stable (Figure 2).

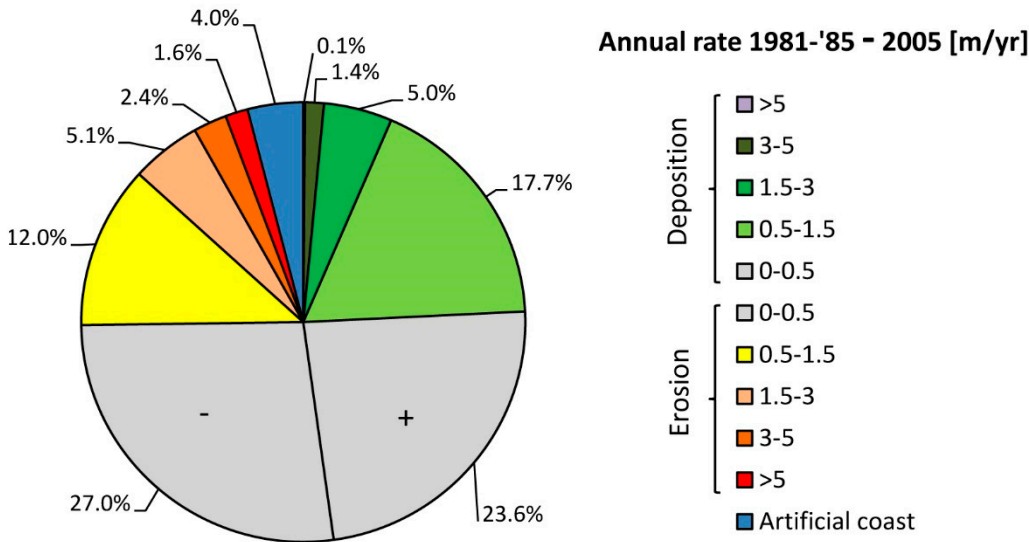

**Figure 2.** Synthesis of the shoreline displacement rate of the Tuscan coast from 1981–1985 to 2005 based on an unpublished report made by the University of Florence for the Tuscany Region.

The main cause of coastal retreat is the drastic reduction in past centuries of sedimentary inputs by rivers flowing onto this coast and the adjacent regions due to reforestation within watersheds, riverbed quarrying and the construction of weirs and dams [42]. This was reflected by the great retreat observed at river deltas, e.g., at the Arno and Ombrone River mouths (Sectors 12, 13, 45 and 46; see Figure 1). On the northern Tuscan coast, the Magra and Serchio river supplies are approximately 632,000 and 23,000 t/yr, respectively [44,45], and the R. Arno, the most important of the region, contributes approximately 1,524,000 t/yr of the total load [46], which is far lower than that of the previous centuries, i.e., only 37% of that estimated for the 1500–1800 AD period [47].

In the central Tuscan coast, the total load of the Cecina River is approximately 250,000, with half of that being bedload. In the south, the Ombrone River total load is estimated to be $1.35 \times 10^6$ m$^3$/yr [45], far lower than the $5 \times 10^6$ m$^3$/yr estimated by [45,48] for the XVI–XVIII centuries. For the other small water courses emptying onto the Tuscany coast, no bedload data are available, but previous authors have considered all of them (except R. Albegna, which has a drainage area of 749 km$^2$) to be insignificant to the sediment input.

A minor volume on the regional scale, but important at the local level, was lost in harbour dredging projects to respect the environmental legislation, which led to sediments being deposited offshore or in a Confined Disposal Facility area. In one case, the port authority was asked to compensate the

dredging operations by nourishing a downdrift area with sediments from land deposits [49]. Shore protection projects, which have been carried out since the end of the 19th century at river mouths but at other places too (Figure 1), locally reduced the sediment deficit, limiting the debits but also favouring the shift of beach erosion to the downdrift coastal sectors [50,51] according to the "domino effect" [52].

Littoral cells, determined via petrographic analyses [53], explain the impact of these works and also the presence of few depositional areas, the most important being the one observed at Marina di Pietrasanta (Sector 7), where sediments from the Magra River meet with those from the Arno River (Figure 1). The southern littoral cell is fed by sediments delivered by Albegna River, emptying onto the Latium coast and draining a volcanic area; the sand here is the darkest of that of all the other regional beaches.

## 3. Materials and Methods

Data acquired by the Department of Earth Sciences of the University of Florence during recent decades were used for the computation of available sediment volumes and the assessment of their suitability for beach nourishment. In this paper, firstly, coastal segments with similar morphologies (pocket beaches, barriers, urban vs. rural, shore protection types) and evolution trends (accreting or eroding) were joined into 56 morphologically-uniform sectors ranging in length from 875 to 6946 m, the former corresponding to a beach south of Marina di Cecina delimited by a groin and a creek mouth (Sector 22), and the latter to the Tombolo di Feniglia (Sector 52), the southern tombolo closing Orbetello lagoon (Figure 1).

Secondly, the sedimentary budget was calculated (Appendix A) for each of the sectors (Figure 1). Bathymetric surveys were carried out in 1997–2005 and extended up to a water depth of 10 m along profiles spaced 250 m apart, or 50 m in correspondence of shore protection structures. Depth accuracy, evaluated on several Sea Control Points [54], was 5 cm. For each sector, volume change was calculated considering a prism with dimensions "l" (sector length), "x" (shoreline displacement), "y" (depth of closure), as in Figure 3.

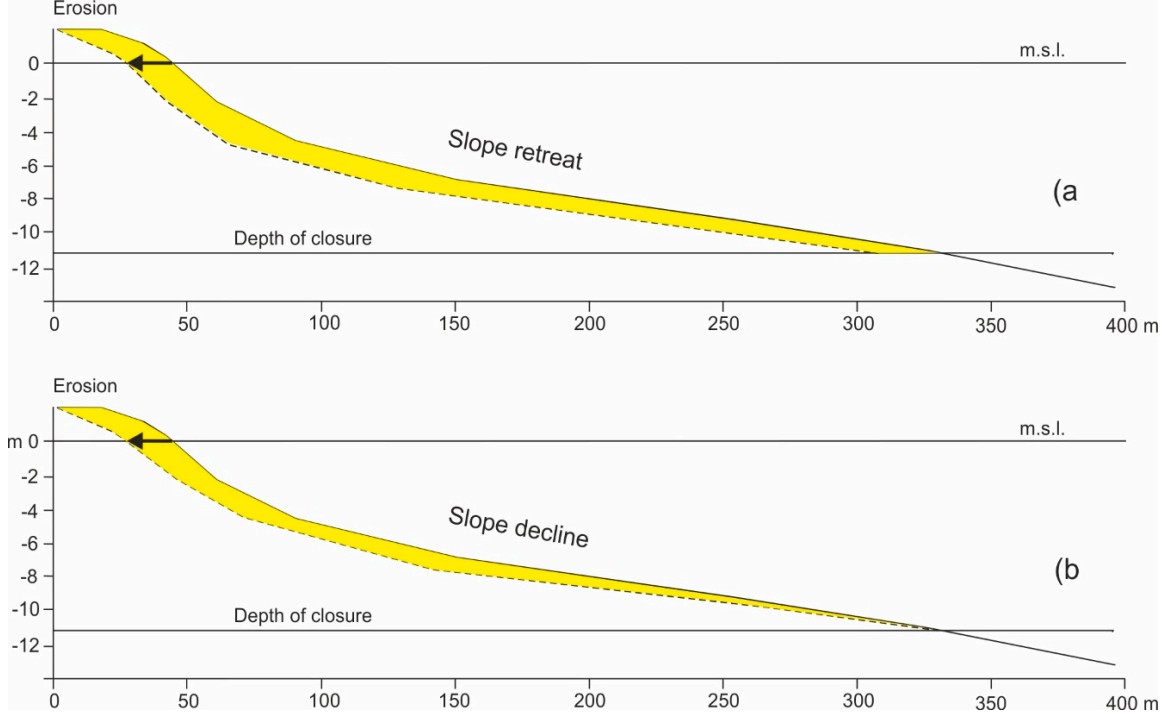

**Figure 3.** Parallel retreat (**a**) and slope declining (**b**) consequent to a beach sediment deficit.

Finally, instead of a profile parallel retreat model (Figure 3a), a slope decline process was considered, i.e., the erosion and accretion along one profile have an area equal to a triangle whose base corresponds with the shoreline displacement and height equal to the depth of the closure (d) obtained by means of the Hallermaier [55,56] equation (Figure 3b):

$$d = 2.28\,H_s - 68.5\,(H_s^2/gT^2) \tag{1}$$

where $H_s$ is the effective wave height just seaward of the breaker zone that is exceeded for 12 h per year, T is the associated wave period and g is the acceleration of gravity. This choice is supported by the assumption that the upper shoreface responds on a much smaller time scale than the lower shoreface, and the idea that the shoreface profile is not always and everywhere in equilibrium with its forcing [57].

For the coast of Northern Tuscany (65 km long), such a model was also supported by [51], who found steeper profiles in accreting areas and milder ones in eroding areas, as dictated by the inability of the depth of closure point to migrate. In addition, a comparison between bathymetric maps in front of the three most important Tuscany river mouths from 1881 and 1976 showed nonsignificant depth changes (i.e., within the accuracy of the method) offshore of −10 m contour depth [47], which is the depth of closure for a return period of a 50 yr storm [55,58]. Obviously, the volumes (positive and negative) are double if a parallel profile retreat model is considered. For the sediments present in the sectors characterized by a well-established sediment surplus, a hypothesis of their use for the nourishment of eroding sectors was elaborated. For these cases, the suitability of sediments in the deposition areas to fill eroding sectors, both in the dry beaches and in the shorefaces, was evaluated for stability and colour compatibility.

Sediment stability was assessed by computing the Stability index (Si) [23], comparing potential borrowed sediments with the native ones of the eroding beaches. The Si is based on the assumption that the stability of each new grain deposited on the beach is inversely proportional to the percentage of its size in the native sediment cumulative distribution. This data also derives from the archive available at the University of Florence of dry beach and shoreface samples collected in several past projects, upgraded with samples collected for this study, all sieved at a $\frac{1}{2}$ phi interval.

For sand colour assessments and compatibility, the CIEL*a*b* quasi-uniform colour space [37] was used, where coordinates are L* (Lightness), a* (Green–Red axes) and b* (Blue–Yellow axes); the perceived distance between two colours is their Euclidean distance in that space. Measurements were performed with a Konica Minolta CR-410, and the acceptability range was that adopted by the Tuscan Region in several nourishment projects, and recently included in the "National Guidelines for Coastal Erosion in Italy" [26], although small variations were allowed, according to the naturalness of each site:

$$-3 < \Delta L^* < +9 \qquad -3 < \Delta a^* < +3 \qquad -1 < \Delta b^* < +3 \qquad \Delta E^*ab < 10 \tag{2}$$

For each of the 56 coastal sectors, a detailed information form was filled out comprising a site description, photos, shoreline evolution, shore protections (if any), sediment budget, previous nourishments (if any), grain size characteristics and colour determination. For specific sites, additional information is given, e.g., core position, dry beach and shoreface sediment texture maps, etc. (Appendix B).

## 4. Results and Discussion

In order to counteract coastal erosion [44], actions to increase river sediment input are needed; these include the favouring of soil erosion and the demolition of dams and weirs, as well as the abandonment of the construction of river expansion tanks. All these actions increase flood risk in inhabited areas, and therefore, are opposed by most stakeholders and, consequently, are not proposed by politicians. Hence, the only viable strategy to balance the negative sediment budget is to increase the credits by artificial beach nourishment. Due to the shortage of shelf sediment (relict gravel and

sand reservoir [55]) to be used for artificial beach nourishment, and environmental limitations for making use of inland quarried materials, the Region of Tuscany is looking with interest at coastal sediments, at least for small projects and seasonal beach profile maintenance works. Hence, dry beach and shoreface sediment deposits along the Tuscan coast have been assessed for their suitability to nourish eroding sectors, considering their stability for the potential beach destination and colour compatibility with local native sand.

Examples of this strategy exist in USA, with sediment bypassing across barrier island inlets [59,60] in Louisiana [61], and in Australia, with the Tweed River Sediment bypassing, which serves to maintain the Gold Coast beaches and related tourist activity [62]. Finally, beach nourishments carried out in Portugal since 1950 have essentially used (>62%) sediment deposited near or inside harbours [63].

Compared with the traditional systems to define Regional Sediments Budgets, the one here developed is based both on the need to maintain the original beach colour (for landscape and environmental reasons [37,38]) and to guarantee the highest possible fill longevity [23] without strongly modifying the beach itself; this is essentially linked to sand size [39].

On the Tuscan coast, assessment of the sediment budget gives an overallannual deficit of 88,452 $m^3$, differently spread within the sediment cells of the coast, with only the sectors fed by the Albegna River (Sectors 49, 50, 51) showing a very small surplus (0.6 $m^3$/m/yr, Figure 4). Identifying the processes driving sediment leak and quantifying related volumes is beyond the scope of the present research, but would be of interest of the Regional Administration, and will be the object of future research. Abrasion is generally considered a minor issue [64], but in mixed sand and gravel sediment beaches, i.e., several sites on the Tuscan coast, it is a significant cause of loss [65]. Landward sediment transport by wind, especially where dunes have been lowered, or vegetation has been cut to create promenades, parking areas and houses, is an effective process of beach sediment loss [12]. On managed beaches, litter and *Posidonia oceanica* leaf removal involves the subtraction of large quantities of sand [66]; this activity is intensively carried out along most of the Tuscany coast. In addition, sand deposited inside harbours is dredged and frequently dumped offshore, as it is not suitable for beach nourishment; a volume of 305,000 $m^3$ of sand and silt was lost in this way at the Marina di Carrara harbour from 1993 to 2008 [49].

In this paper, the volumes to be extracted have been determined at a few locations (Figure 4). In two cases only (Viareggio, Sector 10, and Marina di Pietrasanta, Sector 7, Figure 4), the volumes were sufficient to justify the implementation of a large nourishment project, but in both, the sand is too fine to be stable on beaches that need additional sediment input, as the Si value shows. In other cases, e.g., the updrift of major groins and jetties (e.g., Pietrabianca, Sector 17, and Marina di Grosseto, Sector 43, Figure 4), limited volumes can justify the utilization of only small mobile bypassing system or road transport to address seasonal sand needs on tourist beaches. Some contexts for the possible use of the results of this study in a regional sediment management program are hereafter described.

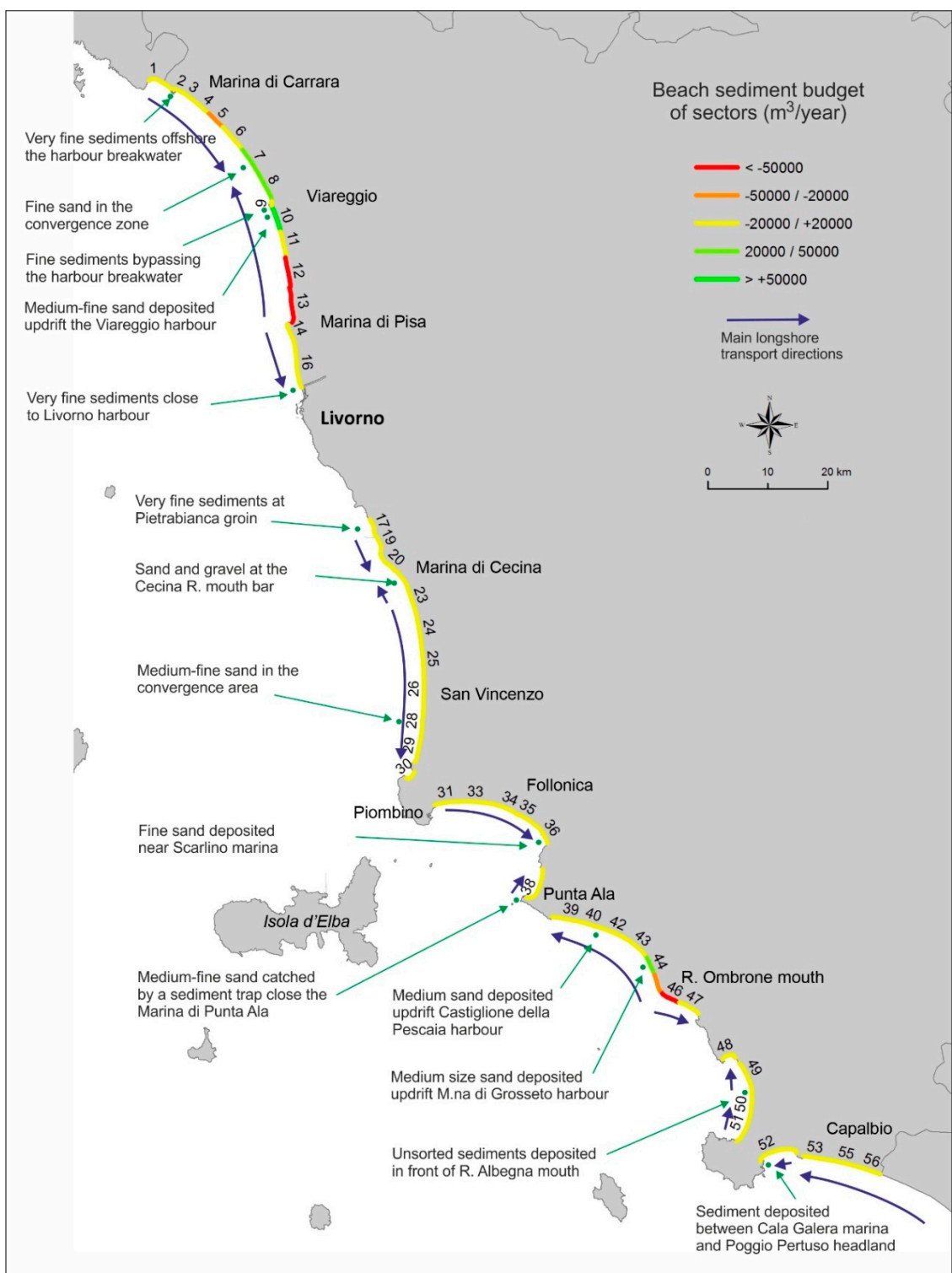

**Figure 4.** Potential dry beach and shoreface borrow sites for beach nourishment.

### 4.1. The Beach Updrift of Viareggio Harbour (Sector 10)

The largest sediment deposit of the Tuscan coast is present south (updrift) of Viareggio harbour (Sector 10; Figures 4 and 5), where the progressive expansion of the jetties emplaced (first) at the entrance of the boathouse, and (later) of the harbour, made the beach expand by approximately 630 m from 1878 to 2005. Today, sand partially blocks the harbour entrance, where a bar makes access for

boats dangerous; many rescue operations are made each year to assist boats that come to a stop on the shoal and, in 2009, there was a fatal accident.

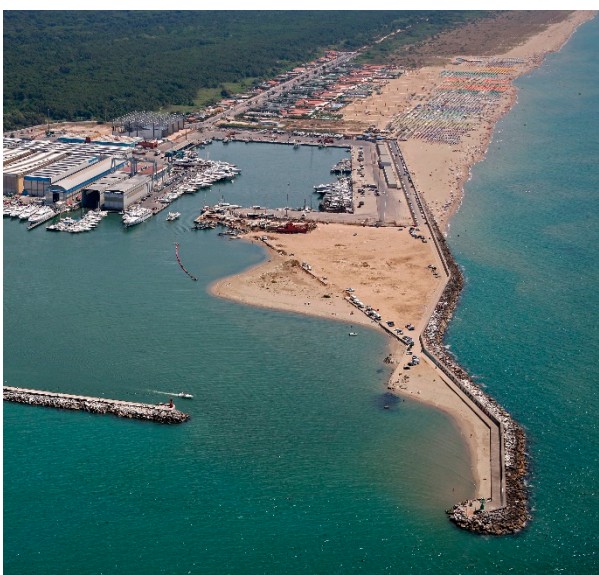

**Figure 5.** The updrift breakwater of Viareggio harbour and the beach close to it. Note that the sun umbrella are far from the cabanas, bars and restaurants.

A volume of approximately 1.5 M cubic meters of sand can be excavated to make the shoreline retreat to the 1986 position, leaving a beach large enough to support the intense tourist activity of this site. Actually, the present beach is too wide for optimal tourist use, as sun umbrellas, positioned near the sea, are too far from the cabanas, bars and toilets (Figure 5) [41]. Excessive sand accumulation updrift affecting ports and structures is a common trend in many Mediterranean beaches as observed in Spain [11,12], France [8] and in the south of Italy [67,68].

Due to the thickness of these deposits, 21 two-meter-long cores were taken along the 4.5 km beach updrift of the harbour (Sector 10) to assess sand characteristics of the layer to be used, which proved to have a greater size and colour homogeneity. Unfortunately, the sediments (borrowed sand in Figure 6a,b) were finer than those present in the beaches that needed nourishment work in this sedimentary cell (e.g., Sectors 5 and 12; see Figures 4 and 6). The beach of Sector 5, at Ronchi (native sand in Figure 6a), is more to the north, but it has no fine sediment because it is fed by sand from the north, i.e., the Magra River, that brings coarser sediments. The Si is 0.342, lower than 0.500, which characterizes borrowed sand of equal grain-size distribution to the native one.

Further, the sediment updrift of the harbour (Sector 12, R. Morto beach) are also too fine (native sand in Figure 6b), with a Si of 0.315, to nourish the area from which they originate, i.e., the Arno River mouth (now mostly from the erosion of the river delta) where the shoreline retreats at 3.56 m/yr, with an unitary annual sediment negative budget of 15.3 $m^3$/m/yr (see Figure 4).

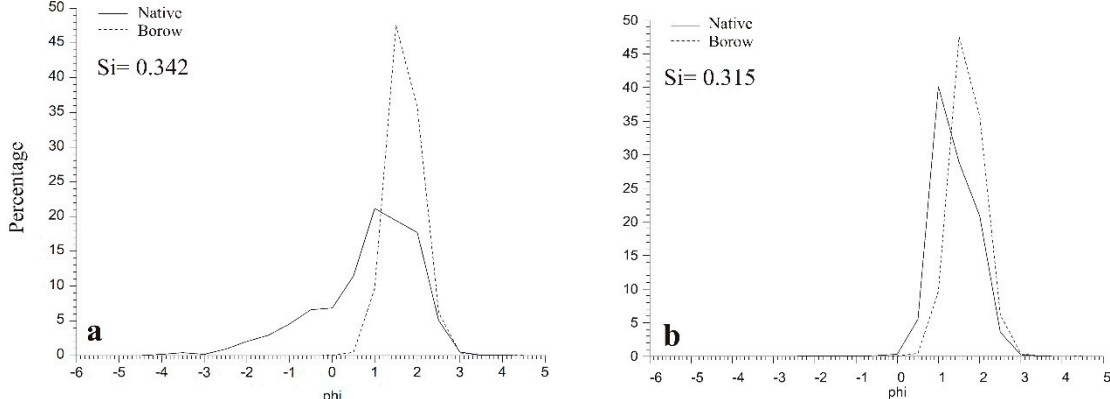

**Figure 6.** Grain size distribution of the composite sample formed with the 14 cores taken on the 4500-meter-long southern Viareggio Sector (10, borrowed sand) and those of the beach of Sector 5 (**a**, native sand at Ronchi) and 12 (**b**, native sand at R. Morto beach).

Concerning colour characteristics, Viareggio sand (Sector 10) is compatible with that of the areas to be nourished. All the parameters (L*, a*, b*) are within the required limits, and dE*ab is respectively 8.71 and 2.19 (Figure 7). The lower colour difference is with Morto River beach (Sector 12), which is part of the same sedimentary subcell fed by Arno River sediment. The larger differences observed with the Ronchi beach (Sector 5), which poses Viareggio sand near the acceptability range, is given by higher L* and b* values; i.e., the sand is lighter and more yellow, both of which meet with appreciation from the stakeholders. The town of Ronchi has an urban beach, extremely anthropized with bathing establishment and several shore protections structures; and a small change in colour would effect an environmental impact. Changes in sand colour as a result of artificial nourishment have been accepted in different ways by stakeholders; at Varadero (Cuba), where beachgoers are not residents, the darkening of the beach created no problems, whereas on the urban beach of Cagliari (Italy), a similar change of colour led to legal action [38,39]. Although considered unstable, the Tuscany Region has already devised a contract for a first project to dredge 100,000 m$^3$ of sand from Viareggio to nourish this beach.

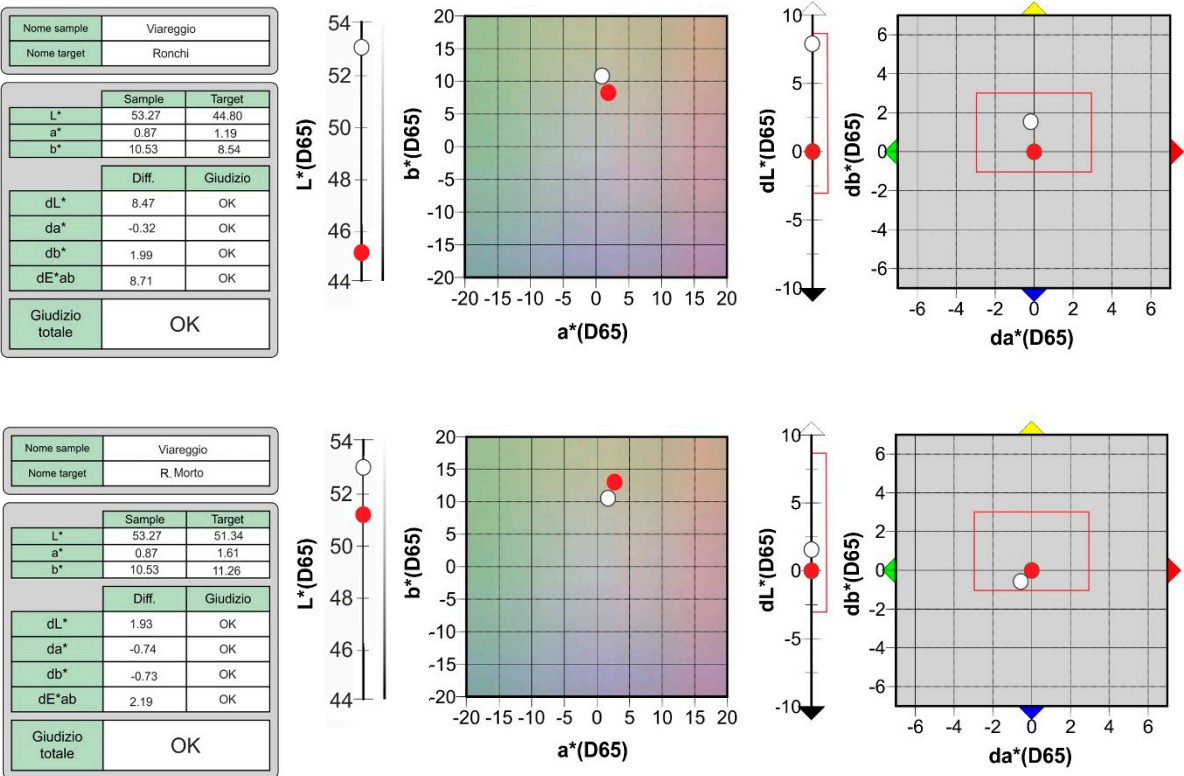

**Figure 7.** Colour compatibility of Viareggio sand with that of the Ronchi beach (Sector 5) and of R. Morto beach (Sector 12). Acceptability range is delimited in red on dL* axis, with a red rectangle on a*, b* plain. Left graph shows L* and a*b* absolute values of native (withe dot) and borrowed (red dot) sediments; the graphs on the right show the distance of borrowed sediments from native ones.

### 4.2. Marina di Grosseto (Sector 44)

Another sediment accumulation point along the regional coast, albeit one that is less important than the former, is the Marina di Grosseto recreational harbour (Figure 8), built at the terminal course of an artificial channel in southern Tuscany. The channel, draining the northern Grosseto plain, was excavated in the 1950s and, to prevent the closure of its mouth, a short jetty was built on the southern side. This structure was later lengthened to cope with the growth of the beach, triggering an asymmetry in the shoreline that, in 1998, was 75 m. Originally, the channel was used as a mooring place for small boats but, in 2000, excavation started for a dock and twin curvilinear jetties to prevent entrance silting and allow safe access. Now, the marina can host boats up to 24 m long.

South of the jetty, the beach further expanded by approximately 25 m along a ca. 3156 m segment; the sediment budget for the investigated period is positive, with +7.1 m$^3$/m/yr, and a mean shoreline displacement of +1.56 m/yr (Figures 1 and 4).

In order to characterize the sediment at Marina di Grosseto, indicated as borrowed sand in Figure 9a,b, 12 cores were taken on the dry beach and on the nearshore bar along a 2 km sector south of the jetty. Their grain size and colour (Figures 9 and 10) were shown to be compatible with those of the two northern, erosive sectors of the sedimentary cell (Sectors 39 and 40) some 10 km away, recording a negative sediment budget of 0.7 and 2.1 m$^3$/yr respectively (native sand in Figure 9a), and/or the beach to the south near the Ombrone River mouth (Sector 45, native sand in Figure 9b), which is retreating at a rate of 2.31 m/yr, with an annual sediment deficit of 10.5 m$^3$/m/yr, but with an erosive hotspot of 10 m/yr near the river mouth (Figures 1 and 4).

However, the final destination of this small sand treasure will be decided within a general program of coastal management of the area that the Region of Tuscany is designing on the basis of the present study. The northern sectors are intensively used by tourism, although the coast has an

ecologically-sensitive area [69] behind a dune belt. During storms, the dune foot is affected by the run-up and eroded, bringing about the loss of several pines. The beach extension could be useful for environmental and economic reasons although, because the low fill stability (Si = 0.240), nourishment should be considered as ephemeral and would require periodic renourishment works to maintain the beach attractive for tourist activity. If this sand is used to nourish a southern area, close to the Ombrone River mouth, no colour differences will be observed and the fill will result much more stable (Si, i.e., 0.525). However, here, the coast is part of the Maremma Regional Park; it is almost uninhabited and the land is a few centimetres above the mean sea level, and locally below the mean sea level. Therefore, a strategic retreat would be the most sustainable solution for this area.

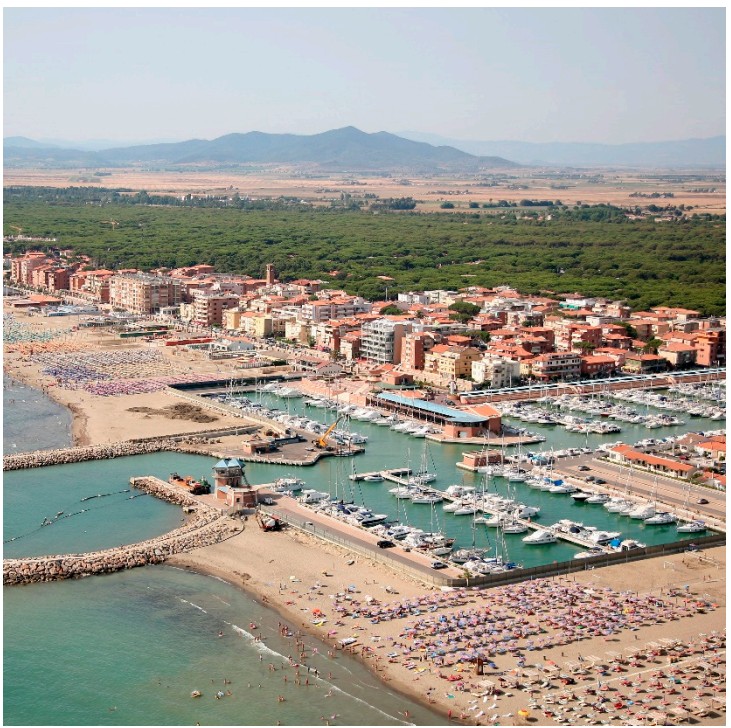

**Figure 8.** The boathouse at Marina di Grosseto, whose entrance is protected by two jetties. The updrift one intercepts the long shore sediment transport, creating an extensive beach.

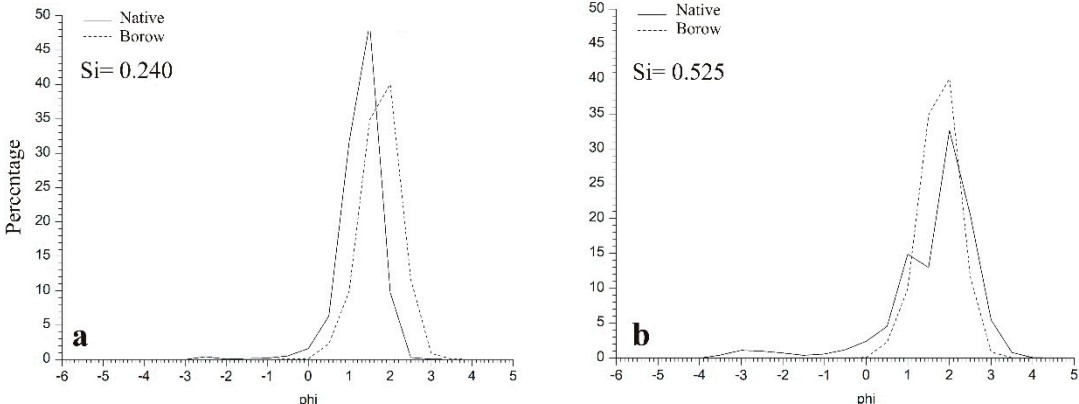

**Figure 9.** Grain size distribution of the composite sample formed with samples of the borrow area, Marina di Grosseto (12 cores of 4 samples each), compared with those of the potential fill beaches to the northern end of the sedimentary cell (Sectors 39 and 40, **a**, native sand at Roccamare beach) and that representing the beach north of the Ombrone River mouth (Sector 45; 8 samples, **b**).

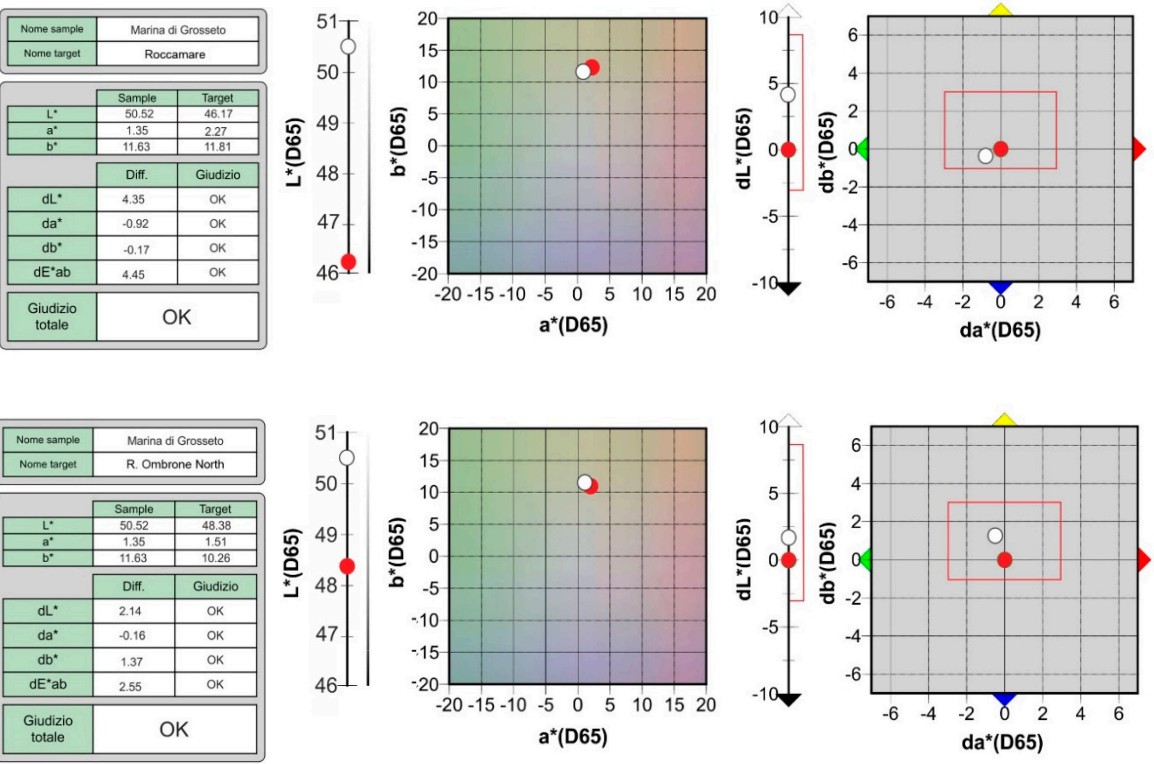

**Figure 10.** Colour compatibility of Marina di Grosseto sand (Sector 44) with that of Roccamare (Sectors 39, 40) and of Ombrone River North (Sector 45). Acceptability range is delimited in red on dL* axis, with a red rectangle on a*, b* plain. The left graph shows L* and a*b* absolute values of native (with dot) and borrowed (red dot) sediments; the right graphs show distance of borrowed sediments from native ones.

## 4.3. Marina di Pietrasanta (Sector 7)

Larger volumes are available in the convergence area at Marina di Pietrasanta [21], where sediments from the north (brought to the coast by the Magra River) meet those coming from the south (from the Arno River); see Figures 1, 4 and 11. Here, the beach has grown by approximately 250 m since 1878, which is the date of the oldest reliable topographic map; the 1984–2005 sediment budget is + 5.7 m$^3$/m/yr, with a beach expansion of +1.3 m/yr [23]. Unfortunately, the natural downdrift fining process make this sand the finest and best sorted of all the 65-km-long coastal cell between the Magra River mouth and Livorno [70]; this is a common trend within littoral cells, and has been described by several authors [10,17,21].

The Si of this sand (Figure 12a), if used to nourish the Ronchi beach (Sector 5), is extremely low (Si = 0.065), whereas in the case of the nourishment of the Morto River beach (Figure 12b), the Si is a bit higher (Si = 0.063) but never so much as to influence the evolutionary trend of this beach, not even in the short term.

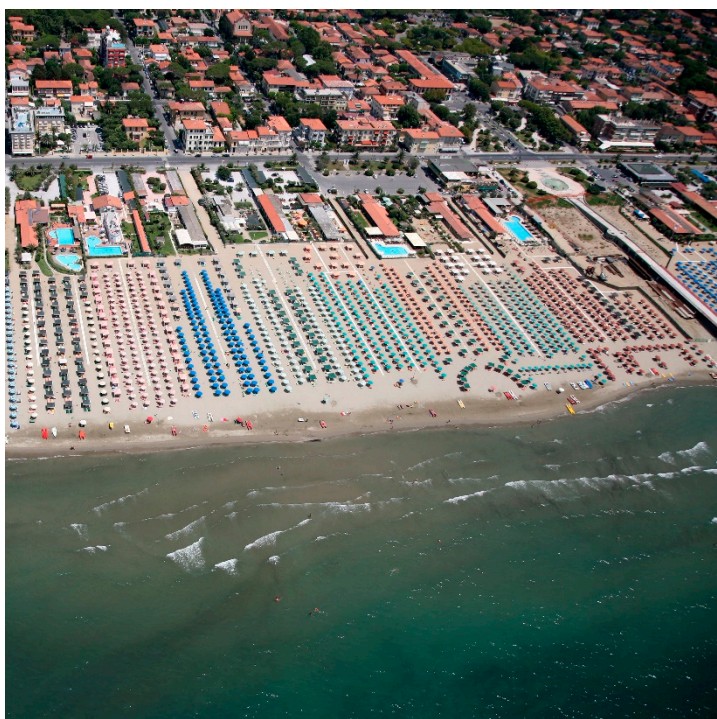

**Figure 11.** The beach at Marina di Pietrasanta, in the sediment transport convergence area.

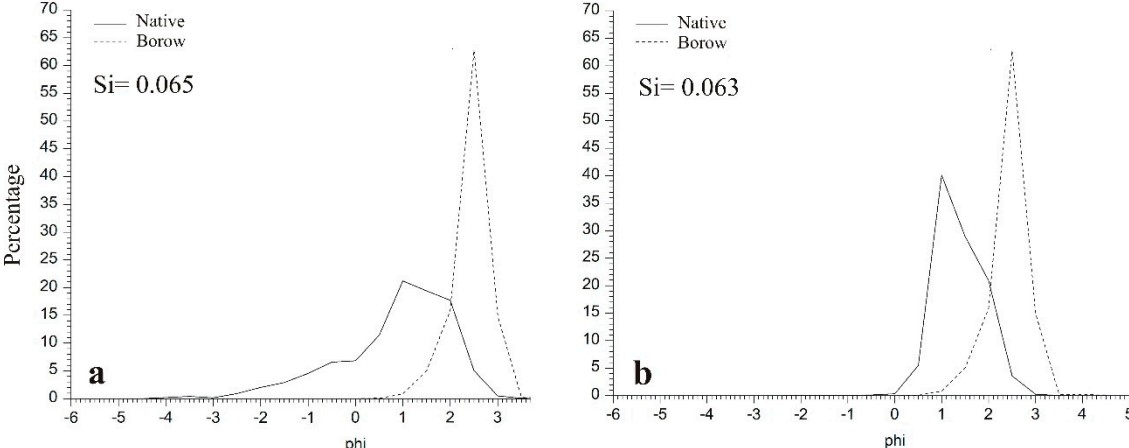

**Figure 12.** Grain size distribution of the composite sample formed with five samples from Pietrasanta beach (Sector 7, borrowed sand in **a**,**b**) and that representing the native beach sand at Ronchi (Sector 5, **a**) and north of the R. Morto beach (Sector 12, **b**), both averaging three samples.

However, at Marina di Pietrasanta, the entire dry beach is intensively used for tourism activities (Figure 11), and it seems economically and politically impossible to accept a reduction of its surface area. At most, an agreement could be reached with local stakeholders to maintain the present beach width, allowing periodical dredging for the benefit of other coastal sectors, where this sand could help to form/enlarge a beach for the tourist season only.

As far as colour is concerned (Figure 13), compatibility is guaranteed at the Morto River (on the southern subcell), but not at Ronchi (northern sub-cell), suggesting an asymmetry in the longshore transport in the convergence zone.

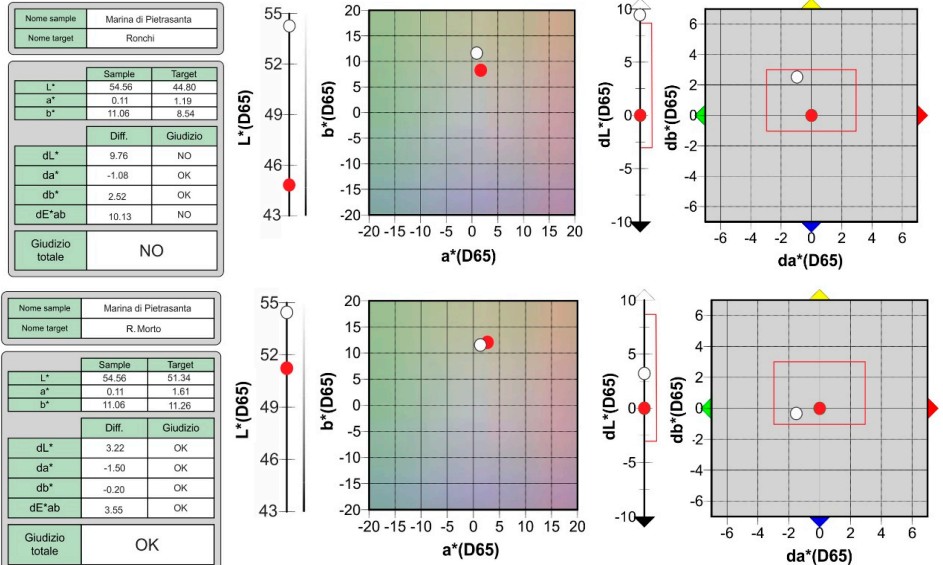

**Figure 13.** Colour compatibility of Marina di Pietrasanta sand with that of Ronchi beach (Sector 5) and of Morto River beach (Sector 12). Acceptability range is delimited in red on dL* axis, with a red rectangle on a*,b* plain. The left graph shows L* and a*b* absolute values of native (withe dot) and borrowed (red dot) sediments; the right graphs show the distance of borrowed sediments from native ones.

## 5. Conclusions

Beach monitoring and maintenance is linked to beach function as part of the defence of low-lying coastal areas and assets for the regional tourism industry. On a regional level, approximately 48% of the studied beaches are eroding, notwithstanding the solid protections (seawall, revetments, detached breakwaters, groins) which have been built since the end of the 19th century. Coastal erosion in this region is the result of a sediment deficit triggered by the reduction of river input for land use changes, dam construction, riverbed quarrying, etc., and no actions are planned in the foreseeable future to solve this issue. Furthermore, sea level rise will increase the deficit in coming years.

Sediment artificial input is the only solution to balance the beach budget; this has been performed since the middle of the 20th century using land-quarried aggregates, mostly on the alluvial plains, with the relevant environmental impact. Research to locate, quantify and characterize Holocene shelf deposits has been carried out by the Region of Tuscany, but suitable sediments have not been found yet. Shoreface deposits were therefore considered, assessing their suitability for beach nourishment in eroding sectors joining the same sedimentary cell. Whilst sediment colour poses limited problems, being in many cases compatible with that of the native beaches, grain size is a major issue, since the sands available in depositional areas are finer than those present on eroding coastal sectors.

Sediments which have accumulated updrift of man-made structures or in natural convergent areas can be used to carry out ephemeral nourishment works only, i.e., to create/enlarge narrow beaches of limited temporal durability, which are able to support beach tourist activities for one summer period, but further investigations are needed to define dredging costs, since mob-demob expenses and the distance of the filling area greatly influence sand cost. In any case, the use of sediments does not solve the structural problem of a lack of sand, because the overall sediment budget of the Tuscan coast is negative ($-88,452$ m$^3$/yr). Offshore loss due to sediment friction cannot be eliminated, but wiser management of dunes and coastal vegetation could reduce wind landward transport.

A further reduction of sand loss linked to human activity could be achieved by carrying out hand beach cleaning operations and avoiding *Posidonia oceanica* removal. Sediment bypasses at harbour entrances could limit the need for harbour dredging, which negatively impacts beach sediment budget. In any case, a regional coastal sediment management plan, like the one proposed in this paper,

constitutes a useful instrument which can assist decision makers and designers in finding cost-effective solutions for shore protection from an integrated coastal zone management perspective.

**Author Contributions:** E.P. coordinated the research, I.C. made the bibliographic research and wrote the Introduction, L.E.C. made the comparison with similar national and international projects, G.A. reviewed the manuscript and wrote the Conclusions. All authors have read and agreed to the published version of the manuscript.

**Funding:** This research was founded by the Regione Toscana (Italy).

**Acknowledgments:** This is a contribution to the PAI Research Group RNM-328 of Andalusia (Spain).

**Conflicts of Interest:** The authors declare no conflict of interest.

## Appendix A

**Table A1.** Sectors in which the Tuscany continental beaches have been divided and related parameters. Investigated time changes from sector to sector depending on the specific year in which the starting survey was performed, always within the 1981–1985 time span.

| Number | Sector Limits | Length (m) | Surface Variation 1981/5–2005 (m$^3$) | Shoreline Displacement 1981/5–2005 (m) | Time (yrs) | Displacement Rate (m/yr) | Depth of Closure (m) | Annual Sediment Budget (m$^3$/yr) | Unitary Annual Sediment Budget (m$^3$/m/yr) |
|---|---|---|---|---|---|---|---|---|---|
| 1 | Magra–Porto di Carrara | 4286 | 8053.0 | 1.9 | 20 | 0.09 | 9.1 | 1832 | 0.4 |
| 2 | Porto di Carrara | | | | | | | | |
| 3 | Porto di Carrara–Marina di Massa | 2,250 | −13,522.5 | −6.0 | 20 | −0.30 | 9.1 | −3076 | −1.4 |
| 4 | Marina di Massa–Foce Frigido | 3000 | 30,471.3 | 10.2 | 20 | 0.51 | 9.1 | 6932 | 2.3 |
| 5 | Foce Figido–Foce Versilia | 3120 | −112,816.1 | −36.2 | 20 | −1.81 | 9.1 | −25,666 | −8.2 |
| 6 | Foce Versilia–Pineta della Versiliana | 5000 | 47,627.1 | 9.5 | 20 | 0.56 | 9.1 | 10,835 | 2.2 |
| 7 | Pineta della Versiliana–Lido di Camaiore | 5000 | 126,213.5 | 25.2 | 20 | 1.28 | 9.1 | 28,714 | 5.7 |
| 8 | Lido di Camaiore–Porto di Viareggio | 5000 | 136,999.3 | 27.4 | 20 | 1.37 | 9.1 | 31,167 | 6.2 |
| 9 | Porto di Viareggio | | | | | | | | |
| 10 | Porto di Viareggio–Marina di Torre del Lago | 4500 | 250,576.8 | 55.7 | 20 | 2.78 | 9.1 | 57,006 | 12.7 |
| 11 | Marina di Torre del Lago–Foce Serchio | 4501 | 59,545.6 | 13.2 | 20 | 0.66 | 9.1 | 13,547 | 3.0 |
| 12 | Foce Serchio–Foce Fiume Morto Nuovo | 5000 | −335,888.8 | −67.2 | 20 | −3.36 | 9.1 | −76,415 | −15.3 |
| 13 | Foce Fiume Morto Nuovo–Foce Arno | 6000 | −390,983.7 | −65.2 | 20 | −3.26 | 9.1 | −88,949 | −14.8 |
| 14 | Marina di Pisa | | | | | | | | |
| 15 | Marina di Pisa–Scogliera Milano | 1715 | −4686.8 | −2.7 | 20 | −0.14 | 9.1 | −1066 | −0.6 |
| 16 | Scogliera Milano–Scolmatore | 6560 | 29,781.1 | 4.5 | 20 | 0.23 | 9.1 | 6775 | 1.0 |
| 17 | Punta Lillatro–Pietrabianca | 2115 | 52,273.4 | 24.7 | 24 | 1.03 | 9.1 | 9910 | 4.7 |
| 18 | Pietrabianca–Pontile Vittorio Veneto | 1080 | −7.036.7 | −6.5 | 24 | −0.27 | 9.1 | −1334 | −1.2 |
| 19 | Pontile Vittorio Veneto–Pontile Bonaposta | 1470 | −16,288.4 | −11.1 | 24 | −0.33 | 9.1 | −3088 | −2.1 |
| 20 | Pontile Bonaposta–Bocca di Cecina | 5419 | −4386.7 | −0.8 | 24 | 0.03 | 9.1 | −832 | −0.2 |
| 21 | Bocca di Cecina–Foce Cecinella | 1500 | 3373.2 | 2.2 | 24 | 0.09 | 9.1 | 640 | 0.4 |
| 22 | Foce Cecinella–Riserva Tombolo di Cecina | 875 | −26,651.9 | −30.5 | 24 | −1.27 | 9.1 | −5053 | −5.8 |

**Table A1.** *Cont.*

| Number | Sector Limits | Length (m) | Surface Variation 1981/5–2005 (m$^3$) | Shoreline Displacement 1981/5–2005 (m) | Time (yrs) | Displacement Rate (m/yr) | Depth of Closure (m) | Annual Sediment Budget (m$^3$/yr) | Unitary Annual Sediment Budget (m$^3$/m/yr) |
|---|---|---|---|---|---|---|---|---|---|
| 23 | Riserva Tombolo di Cecina–Marina di Bibbona | 5544 | −101,043.4 | −18.2 | 24 | −0.76 | 9.1 | −19,156 | −3.5 |
| 24 | Marina di Bibbona–Foce Fosso ai Molini | 5250 | −24,682.5 | −4.7 | 24 | −0.20 | 9.1 | −4679 | −0.9 |
| 25 | Foce Fosso ai Molini –La Riconiata | 5000 | 73,514.5 | 14.7 | 24 | 0.61 | 9.1 | 13,937 | 2.8 |
| 26 | La Riconiata–Porto di S.Vincenzo | 4750 | 7642.7 | 1.6 | 24 | 0.07 | 9.1 | 1449 | 0.3 |
| 27 | Porto di S.Vincenzo | | | | | | | | |
| 28 | Porto di S.Vincenzo–Rimigliano | 4750 | −1893.5 | −0.4 | 24 | −0.02 | 9.1 | −359 | −0.1 |
| 29 | Rimigliano–Torre Nuova | 4620 | 753.3 | 0.2 | 24 | 0.01 | 9.1 | 143 | 0.0 |
| 30 | Golfo di Baratti | 2070 | −10,700.5 | −5.2 | 21 | −0.25 | 7.8 | −1987 | −0.96 |
| 31 | Foce Cornia Vecchia–Torre del Sale | 3214 | −17,065.0 | −5.3 | 21 | −0.25 | 9.1 | −3697 | −1.2 |
| 32 | Torre del Sale | | | | | | | | |
| 33 | Torre del Sale–Foce Fosso Corniaccia | 4500 | −47,588.8 | −10.6 | 21 | −0.50 | 9.1 | −10,311 | −2.3 |
| 34 | Foce Fosso Corniaccia–Canale Allacciante Corvia | 4340 | −1250.4 | −0.3 | 21 | −0.01 | 9.1 | −271 | −0.1 |
| 35 | Canale Allacciante Corvia–Pineta di Levante | 5618 | 15,304.7 | 2.7 | 21 | 0.13 | 9.1 | 3316 | 0.6 |
| 36 | Pineta di Levante–Porto di Scarlino | 2800 | 31,801.6 | 11.4 | 21 | 0.54 | 6.7 | 5073 | 1.8 |
| 37 | Cala Le Donne–Piastrone | 2860 | 4523.8 | 1.6 | 22 | 0.07 | 7.9 | 812 | 0.3 |
| 38 | Piastrone–Punta Hidalgo | 3120 | −31,281.3 | −10.0 | 22 | −0.46 | 6.1 | −4337 | −1.4 |
| 39 | Punta delle Rocchette–Punta Capezzolo | 5750 | −56,002.1 | −9.7 | 21 | −0.46 | 9.1 | −12,134 | −2.1 |
| 40 | Punta Capezzolo–Foce Bruna | 1380 | −4578.4 | −3.3 | 21 | −0.16 | 9.1 | −992 | −0.7 |
| 41 | Foce Bruna–Limite sud abitato Castiglione | 1031 | 13,336.6 | 12.9 | 21 | 0.62 | 9.1 | 2890 | 2.8 |
| 42 | Limite sud abitato Castiglione–Pineta del Tombolo | 4250 | 45,594.4 | 10.7 | 21 | 0.51 | 9.1 | 9879 | 2.3 |
| 43 | Pineta del Tombolo–Porto Marina di Grosseto | 4533 | 75,736.2 | 16.7 | 21 | 0.80 | 9.1 | 16,410 | 3.6 |
| 44 | Porto Marina di Grosseto–Chiaro del Porciatti | 3166 | 103,897.8 | 32.8 | 21 | 1.56 | 9.1 | 22,511 | 7.1 |
| 45 | Chiaro del Porciatti–Bocca d'Ombrone | 3250 | −165.039.7 | −50.8 | 22 | −2.31 | 9.1 | −34,133 | −10.5 |

**Table A1.** *Cont.*

| Number | Sector Limits | Length (m) | Surface Variation 1981/5–2005 (m³) | Shoreline Displacement 1981/5–2005 (m) | Time (yrs) | Displacement Rate (m/yr) | Depth of Closure (m) | Annual Sediment Budget (m³/yr) | Unitary Annual Sediment Budget (m³/m/yr) |
|---|---|---|---|---|---|---|---|---|---|
| 46 | Bocca d'Ombrone–Spiaggia di Alberese | 3250 | −253,503.3 | −78.0 | 22 | −3.55 | 9.1 | −52,429 | −16.1 |
| 47 | Spiaggia di Alberese–Cala Rossa | 3750 | 88,011.7 | 23.5 | 22 | 1.07 | 9.1 | 18,202 | 4.9 |
| 48 | Golfo di Talamone | 2465 | −12,796.0 | −5.2 | 21 | −0.25 | 3.8 | −1158 | −0.5 |
| 49 | Foce Osa–Bocca d'Albegna | 5750 | −2767.1 | −0.5 | 21 | −0.02 | 8.3 | −547 | −0.1 |
| 50 | Bocca d'Albegna–Tombolo della Giannella | 4000 | 11,972.2 | 3.0 | 21 | 0.14 | 8.3 | 2366 | 0.6 |
| 51 | Tombolo della Giannella–Santa Liberata | 4250 | 33,376.1 | 7.9 | 21 | 0.37 | 8.3 | 6596 | 1.6 |
| 52 | Tombolo di Feniglia | 6946 | 56,831.4 | 8.2 | 21 | 0.39 | 8.7 | 11,772 | 1.7 |
| 53 | Torre Tagliata–Palude di Tagliata | 3250 | −29,592.3 | −9.1 | 21 | −0.43 | 8.7 | −6130 | −1.9 |
| 54 | Palude di Tagliata–Macchiatonda | 1500 | −33,376.3 | −22.3 | 21 | −1.06 | 8.7 | −6914 | −4.6 |
| 55 | Macchiatonda–Palude di Burano | 4250 | −40,714.4 | −9.6 | 21 | −0.46 | 8.7 | −8434 | −2.0 |
| 56 | Palude di Burano–Foce Chiarone | 4250 | 9558.7 | 2.2 | 21 | 0.11 | 8.7 | 1980 | 0.5 |

## Appendix B

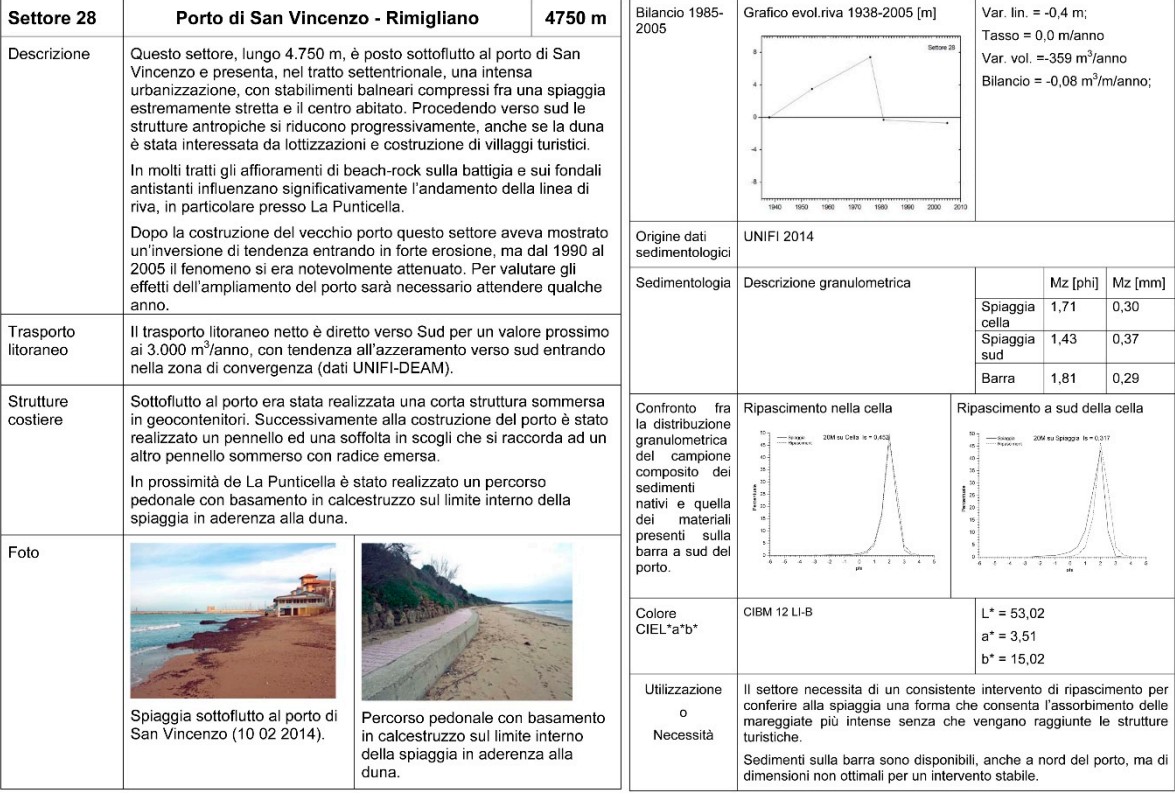

**Figure A1.** One out of the 56 forms (this is in two pages) reporting information on the sectors in which the Tuscan coast was divided to assess sediment budget, sand characteristics and potentials/needs for artificial nourishment (in Italian, like the original).

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
