# Peer review of "An Integrated Coastal Sediment Management Plan: The Example of the Tuscany Region (Italy)"

_jmse, doi:10.3390/jmse8010033_

Round 1

Reviewer 1 Report

See attached file.

Reviewer 2 Report

The presented study is a regional coastal sediment management plan; constitutes an effective solution in coastal protection with an integrated coastal zone management perspective. The study has quantified and characterized in a correct and scientific way sediment availability in the coastal Tuscany area (Italy), consisting mainly of sand accumulated updrift of major ports and marinas and which could allow for significant bypass interventions to nourish eroding sectors.
Beach nourishment is a soft engineering alternative to hard structures on the shore. The hard structures often have negative effects on the downstream, while the artificial beach has no such effect. Beach nourishment is the best alternative to increase the resilience of coastal systems without loss of ecological functions or capacity to provide recreational and other services

Author Response

Thank you very much for your comments. 

Reviewer 3 Report

Review for “An integrated Coastal Sediment Management Plan: the example of the Tuscany Region (Italy)”, Pranzini et al.

The paper presents a sediment budget for the Tuscany region, based on multiple years of data collection, building on previous research outputs. The primary aim is to assess several borrow site / nourishment site options based on colour and grain size suitability. The paper is well written, apart from minor grammatical issues. The figures are suitable. However, I felt more effort should have been made to contextualise the results within the literature and discuss broader implications. Apart from this, I have suggested only minor modifications. If these issues can be attended to, I recommend the paper as suitable for publication in JMSE.

General issues

Grammar / writing style: This is not a major issue, but there are several minor grammatical problems throughout. I highlight some but not all of these in the minor comments below. I suggest the authors go over the manuscript themselves and revise where necessary.

Broader scientific themes: I found the manuscript at times read like a consultancy report, without substantial attempt to address the broader scientific implications of the work. At a minimum, I would suggest: (i) a statement at the end of the abstract contextualising the research; and (ii) and a minimum of 1-2 paragraphs of discussion that place this research within the existing literature and address the novel implications of this work for similar sites. Currently, there has been no serious attempt at a discussion section.

Units: Very minor… Superscript for “m^3”. Number format should be “500,000” or “500 000”, not “500.000”.

Minor points

L12. Grammar, suggest “with a main aim of establishing…”.

L13. Delete “the” in “time span from the 1981…”.

L14. “km” instead of “kilometres”.

L15. Undefined variables: “CIEL*a*b”. Suggest leaving the variables out of the abstract.

L16. Grammar, suggest “… range conforming to …”.

L17. Grammar, suggest “… possibility of using sediment …”.

L19. Grammar, suggest “… volumes are of sufficient magnitude to support…”.

L36. Provide references for “… is the most used approach…”.

L71. Suggest “significant” rather than “relevant”.

L91. More information is required on the methods used to obtain shorelines, without the reader having to refer to [17]. This should include: (i) how were the beach areas obtained? E.g., aerial photos?; (ii) sampling frequency and total duration of the area/shoreline timeseries. The abstract says “1981-2005” and Appendix B appears to show a fuzzy timeseries, but that information should be clearly stated here (or in the methods).

FIG 1. Could a wave rose be added? This helps the reader interpret the longshore transport. Also, could bathy contours be added?

FIG 1. How were the longshore transport arrows determined? Are they indicative/inferred or modelled? Are there approximate magnitudes and uncertainties for the longshore transport?

FIG 2. Has this figure / data been previously presented elsewhere? If so, mention “modified from” in the caption. If not, it seems like results and should be moved there. Further, if the detailed breakdown of erosion/accretion is being presented, then the total budget should be presented at the same time. Then the reader can interpret what the system as a whole is doing as well as variation within the system.

L103. Provide a bit more information on the reduction of river inputs. Is this due to dams that were built many decades ago? What time-scale are we looking at?

L117. I was looking for a more comprehensive assessment/summation of the known major processes acting on the sediment budget. My interpretation based on what is presented is that this section of coast is supplied sediment by several rivers, which has drastically decreased (over an unstated timescale). It then appears that wave-forced (?) longshore transport acts to redistribute sediment within the compartment. Are there any alongshore inputs/outputs from this coastal compartment as a whole? (the south looks open in Fig. 1). Are there any relevant cross-shore processes thought to result in gains/losses? Are tidal processes a factor, and what is the tidal regime? Is there relatively unhindered bypassing/sediment sharing throughout the compartment? Or are there points of low/zero transport (e.g., Piombino in Fig. 1 looks like it may block all transport).

STUDY AREA (general). Little to no information is provided on wave climate. Please add this and describe how wave forcing is relevant to the various sediment transport processes occurring within the cell.

L131-141. and FIG 3. This method seems reasonable and is adequately justified. But this method requires bathymetry, correct? What bathymetric data were used and how were the data obtained? Please provide this information is the methods.

L153, L157. “CIEL*a*b”. Please provide a full description of this method and define all variables, suitable for a reader who has not encountered this method previously.

L178-181. This feels like introduction material?

L182. Now the total budget is presented (-88,452 m^3/year). If this is a previously presented value, it should have been presented along with FIG 2 (and associated text) in the site description. If it is novel, it belongs in the results, but then FIG 2 must also be moved to results. Further, this figure is presented with 5 significant figures, which likely overstates precision, and no uncertainty bounds are provided. I am interested to know where this 9x10^4 m^3/yr is going? Is it likely lost alongshore or offshore?

L191-192. Join the last 1 sentence paragraph to the previous paragraph.

L199. “Overpassing” or “bypassing”? Overpassing is generally used for sand dunes migrating over the top of headlands.

FIG 4. Is it possible to add longshore transport arrows, similar to Fig. 1? I found I had to refer back to Fig. 1 multiple times to aid interpretation.

FIG 7, 10, 13. What are the green, white and red circles? As stated previously, this method should have been explained in more detail in the methods.

L259-282. Several minor grammatical issues in this section, please read over and revise.

L274. Delete “sensu”.

L285. What sector in Fig. 4 is Marina di Pietrasanta in? There are several other instances where sites mentioned in the text could be given a sector number to make it easier for the reader.

RESULTS and DISCUSSION (general). I found the last sections of the results (from L246 onward) somewhat repetitive. It would be possible (just a suggestion, not essential) to combine all 3 sites (FIGS 5,8,11) into a single concise section, by combing similar figures (photos, grain size, grain colour) and synthesising the text. This would save space for a more thorough discussion (see General Issue, “Broader scientific themes”).

L314. Suggest “Coastal erosion in this region is …”.

L316. Change to “… this deficit in coming years.”

L319. Change “Researches” to “Research”.

L327. Delete second use of “only”.

L332. Again, on the annual total sediment budget, where is this loss going to?

Appendix A. Header, typo “Lenght”.

Round 2

Reviewer 1 Report

Reviewer’s remarks on revised manuscript JMSE-668515
Having read the revised manuscript and the authors’ response, I believe that the authors did put an
honest effort into addressing the issues raised during my first review, which have lead to a relatively
“stronger” submission overall. My recommendation is to accept the manuscript for publication in JMSE,
pending a couple minor revisions, as noted in the following remarks (with first review- numbering); these
revisions can be overseen by the handling Editor.
[R.01] Introduction // A couple of references on modelling attempts were added; however, they’re
rather general and not recent. It would be favourable for the completeness of this work if some
additional effort was put on this part (e.g. reference to case studies based on advanced models, like the
ones mentioned in the already included reference of Capobianco et al. 2002).
[R.03] Materials and Methods // It is still not very clear which data were used for the DoC calculation for
each segment. Is it to assume that they relate to the data referred to in the revised Figure 1? Clarification
needed.
1

Author Response

Dear Reviewer

thank you for your suggestions, we added a couple of references

Stive, M.J.F.; de Schipper, M.A.; Luijendijk, A.P.; Aarninkhof, S.; van Gelder-Maas, C.; van Thiel de Vries, J.S.M.; de Vries, S.; Henriquez, M.; Marx, S.; Ranasinghe, R. A New Alternative to Saving Our Beaches from Sea-Level Rise: The Sand Engine. Journal of Coastal Research 2013, 29 (5), 1001-1008. Bird, E.; Lewis, N. Beach Renourishment. Springer, Dordrecht, 2015. p. 137.

and this sentence:

Depth accuracy, evaluated on several Sea Control Points [52], was 5 cm. For each sector volume change was calculated considering a prism with dimensions l (sector length), x (shoreline displacement), y (depth of closure) as in Fig.3.

Thank you very much for your work and efforts. I attach a NEW VERSION of the manuscript.

Best regards, Giorgio Anfuso
